# Four-Dimensional Investigation of Gravel Beach Ridge Accretion and 50 Years of Beach Recharge at Dungeness, UK, Using Historic Images, GPR and Lidar (HIGL)

**Charlie S. Bristow** [1,*] ⓘ**, Lucy Buck** [1] **and Maria Inggrid** [2]

[1] Department of Earth and Planetary Sciences, Birkbeck, University of London, London WC1E 7HX, UK; lucy.buck.15@ucl.ac.uk

[2] Asian School of the Environment, Nanyang Technical University, Singapore 639798, Singapore; MARIAIRE001@e.ntu.sg

[*] Correspondence: c.bristow@ucl.ac.uk

**Abstract:** Dungeness is a cuspate foreland on the south coast of England that is the largest shingle feature in Europe and includes hundreds of beach ridges. It is also the location of two nuclear power stations that were constructed in the 1960s. The dominant southwest waves cause longshore drift from west to east, eroding the southwest side of Dungeness, accompanied by accretion on the east side. A record of this eastward movement and sediment accretion is preserved by the shingle beach ridges. The power stations are located on the eroding southwestern side of the ness, and a system of beach recharge has been used to move shingle from the downdrift, east-facing shore to the updrift, southwest-facing shore to protect the power stations from coastal erosion. We use a novel combination of historic images, ground-penetrating radar (GPR), and Lidar (HIGL) to investigate accretion and beach ridges at Dungeness during the past 80 years. We report changes in accretion along the coast and use GPR to determine the thickness of beach gravels. The amount of accretion, represented by the width of the backshore, decreases downdrift from south to north. The number of beach ridges preserved also decreases from south to north. By combining the shingle thickness from GPR with elevation data from Lidar surveys and records of beach accretion measured from aerial images, we estimate the volume and mass of gravel that has accumulated at Dungeness. Historic rates of beach accretion are similar to recent rates, suggesting that the 55 years of beach recharge have had little impact on the longer-term accretion downdrift.

**Keywords:** shingle; beach recharge; beach nourishment; sediment budget; lidar; ground-penetrating radar

## 1. Introduction

Dungeness is a prominent cuspate foreland on the south coast of England that is around 25 km wide and projects almost 10 km into the English Channel (Figure 1). It is reported to be the largest coastal shingle feature in Europe [1]. The shape of the foreland has evolved through the late Holocene, as it has prograded southwards into the Channel [2]. Shingle ridges on the surface of the foreland record these changes in the morphology of the ness and have been mapped by [3] and [4]. The positions of the shorelines at Dungeness, as shown by historic maps from 1617 and 1800, are used to assess historic changes in the shoreline [3]. The 1617 map drawn by Poker was used by [3] to represent the shoreline in 1600, although, as they note, it is only approximately correct because the shoreline does not correspond with the orientation of the shingle ridges [3]. Lewis and Balchin [3] comment that none of the maps from the next two centuries inspire enough confidence to allow them to draw a shoreline between 1600 and 1800. Subsequent shoreline locations are more accurately represented by Ordnance Survey maps. Measurements from 1871 and 1908 Ordnance Survey maps, supplemented by field surveys in 1938–1939, are included in [3], although there is some confusion over the date of the earlier survey, which is variously

shown as 1871, 1873, and 1878 ([3], p. 268). Their results show that average rates of progradation between 1873 and 1938 decreased from south to north with an average of 4 m/year at the Ness, decreasing to 0.25 m/year around 5.7 km to the north near Greatstone. Much of the shingle at Dungeness accumulated as recurved spits [4]. The change from recurved spit to cuspate foreland morphology at Dungeness is relatively recent [4], and cartographic evidence shows the closure of an inlet between Greatstone and Littlestone between 1879 and 1905, which was probably artificial [4].

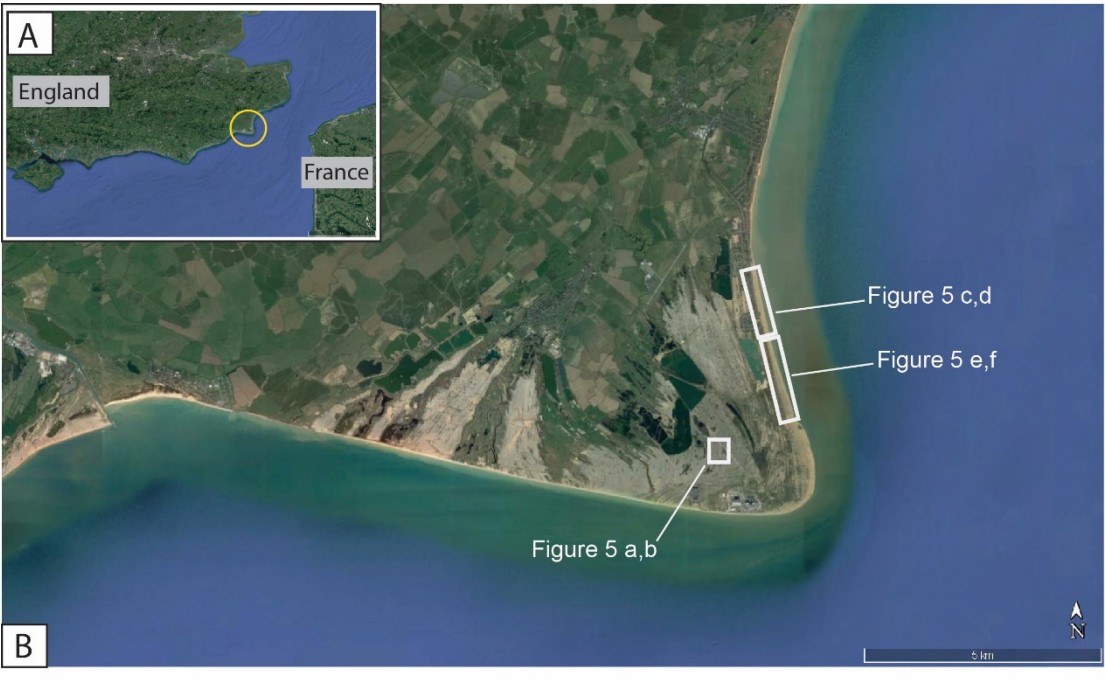

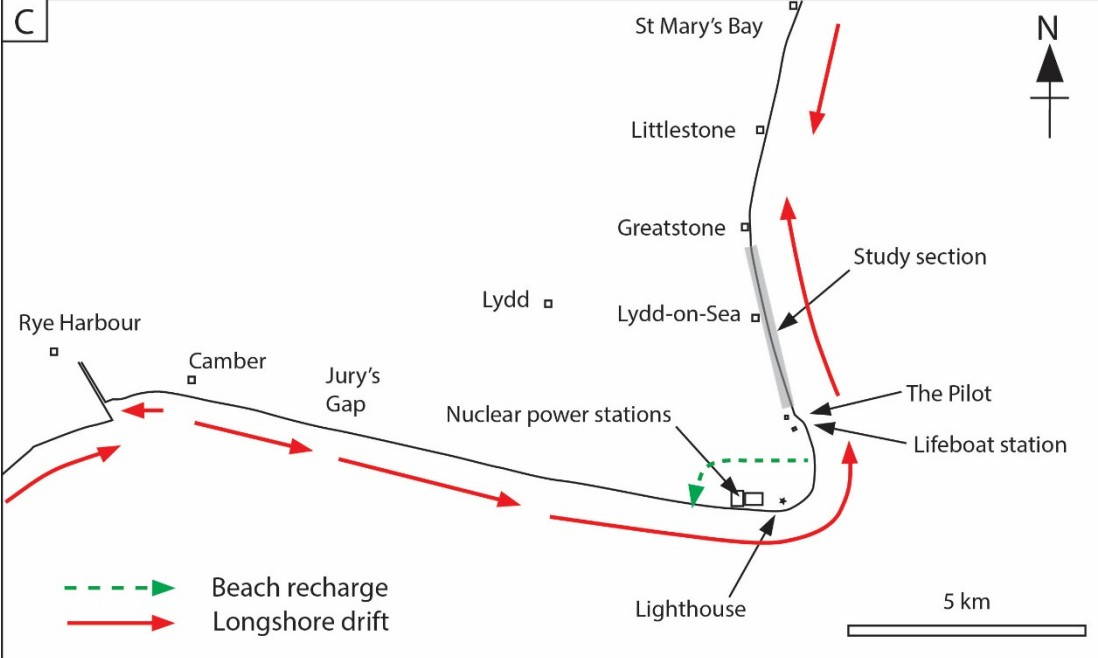

**Figure 1.** (**A**) Dungeness location on the south coast of England. (**B**) Google Earth™ satellite image of Dungeness; the light toned areas are shingle with little vegetation, inset boxes show locations of Figure 5a–f. (**C**) Annotated map showing the shape of the cuspate foreland, locations described in this paper, and the directions of longshore drift (red arrows) from [4]. The study section shown in detail in Figure 3 is depicted by gray tone.

The shingle ridges at Dungeness are beach ridges. The term beach ridge is widely used in the literature and has been applied to different coastal features as in [5,6]. In this paper we follow the definition of [7,8], where beach ridges are defined as 'swash aligned, swash and storm wave-built deposits or ridges formed primarily of sand, pebbles, cobbles (gravel) or boulders, or a combination of these sediments' [7,8] (p.73). Indeed, ref. [8] notes that 'the classic beach ridges are the storm-built shingle and cobble ridge such as those at Dungeness [8] (p. 73). The shingle ridges overlie intertidal and subtidal sands that have OSL ages between 5000 and 400 years [9], indicating that the cuspate foreland is Holocene in age. In the field, there is a sharp break between the shingle foreshore and underlying sand (Figure 2), making Dungeness a composite gravel beach as defined by [10]. The combination of change in slope and change in sediment permits mapping of the sand shingle contact.

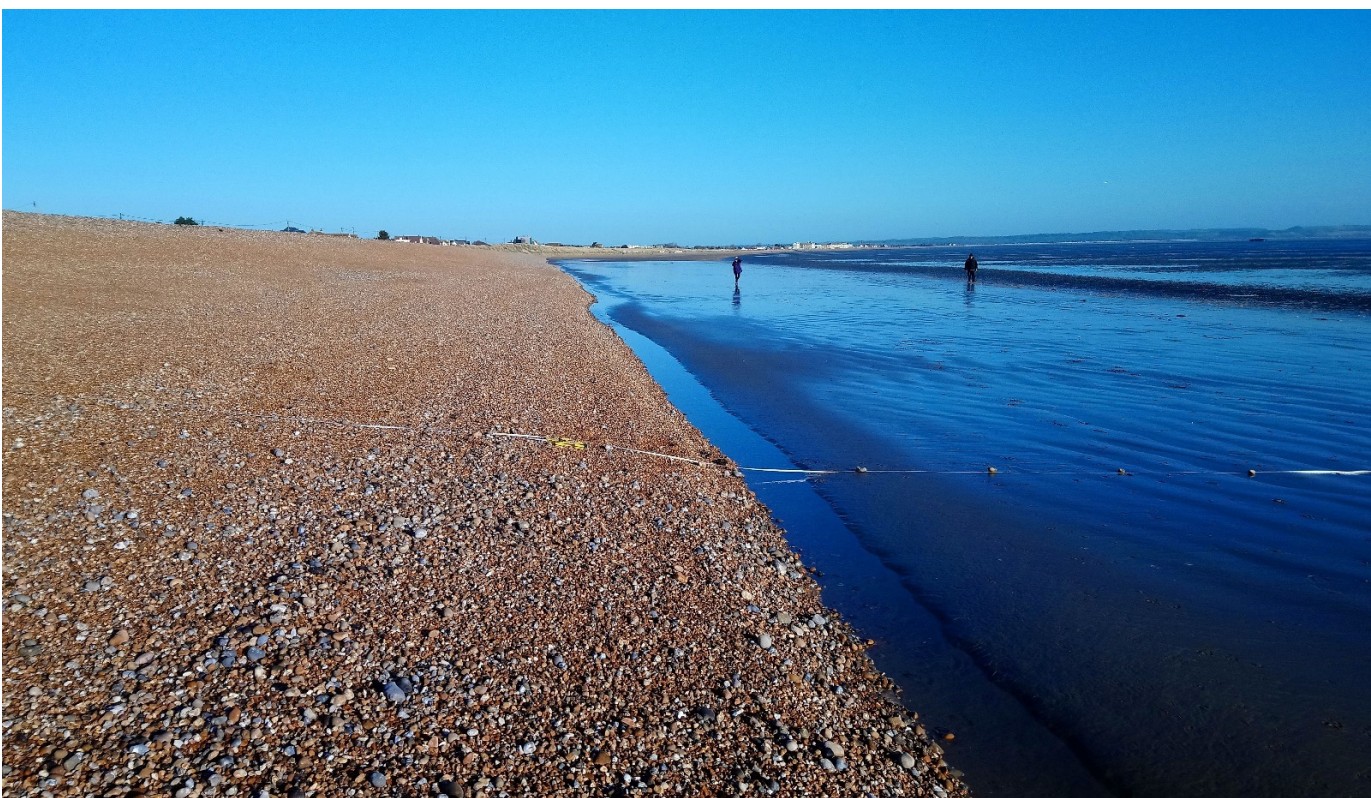

**Figure 2.** Field photograph of shingle sand contact on the beach at Dungeness. Note the sharp nature of the contact, the contrast in tone between darker sand and pale shingle that can be distinguished on aerial images and used to map historic 'shorelines,' as well as the break in slope between the steeper shingle beach and lower angle intertidal sands. The steeply dipping beachface is visible on GPR profiles.

The tidal range at Dungeness is macrotidal with a mean tide range of around 5.3 m. The wave climate is bimodal and bidirectional with the most frequent and the largest waves coming from the southwest and less frequent and smaller waves from the east [11]. The resulting longshore drift causes a net migration of shingle from West to East, accompanied by erosion on the south side of the ness and accretion on the east side [1,2,4]. Historic rates of erosion on the south side, derived from maps dated 1816 to 1906, are similar to the accretion rate over the same period, although the accretion rate may be underestimated as not all of the coast was covered, and there could be inaccuracies on older maps [1]. After 1906, [1] notes that 'there appears to be more erosion than accretion and the volumes moved are decreasing as is the supply of shingle from the west due to coastal works updrift and the development harbour mouth works in Rye' [1] (p. 4).

Dungeness is protected by a range of conservation designations, including National Nature Reserve (NNR), Special Protection Area (SPA), Special Area of Conservation (SAC), and forms part of the Dungeness, Romney Marsh, and Rye Bay Site of Special Scientific Interest (SSSI). However, anthropogenic impact at Dungeness is considerable, and much of this has conspired to deface or remove the beach ridge structures for which the site is famous. Part of the gravel abstracted was used for beach recharge. It is reported that beach recharge started updrift from Dungeness along the coast between Pett and Rye in 1934 [4]. Additional recharge at Jury's Gap, which is on the updrift, south side of Dungeness (Figure 1), started in the mid-1950s and at St Mary's Bay on the East side of Dungeness in 1978 [4]. The volumes of shingle added to beaches in 1979 were Pett Wall, 29,301 $m^3$, Jury's Gap 16,719 $m^3$, Power Station 26,697 $m^3$, and St Mary's Bay 37,834 $m^3$ [4].

The activities that relate directly to this study include beach recharge to protect the nuclear power stations at Dungeness and maintaining access to launch the lifeboat. There are two nuclear power stations at Dungeness, Dungeness A and Dungeness B, both located on the southwest facing coast which is eroding. Construction of Dungeness A started in 1960, and construction of Dungeness B started in 1966. Part of the sea defenses for the power stations includes beach recharge, which has been running continuously since 1965 [1]. Initial volumes of shingle were 15,000 $m^3$/year increasing to 20,000 $m^3$/year and up to 70,000 $m^3$/year in 1992 [1]. A change in recharge practice in 1992 led to a reduction in the amount of shingle moved, and [12] reports that 29,000 $m^3$ was removed from the beach for recharge. In addition, there is almost daily movement of shingle by bulldozer at the lifeboat station to maintain a slipway to launch and recover the lifeboat. The road along the coast was built in 1935, and during the second world war, 1939–1945, there were anti-tank and anti-personnel devices constructed on the beach because Dungeness was seen as a potential landing site for invasion.

GPR is widely used to image the strata preserved within beaches and coastal plains in 2-D and 3-D, e.g., [13–17]. In addition, GPR has been combined with optical dating to reconstruct rates of beach progradation and sediment accretion to investigate histories of sea-level and storminess, e.g., [18–22], as well as sediment budgets [23].

In this paper we study a 3.5 km section of the coast between The Pilot public house in the south and the village of Greatstone in the north (Figure 1). This section was selected because it is relatively free from anthropogenic influence, with the southernmost GPR line (Beach 1) located 500 m north of the lifeboat station, and 1855 m downdrift from the lighthouse (Figure 1). The section studied forms parts of sub cells RS4 and RS5 of the Strategic Regional Coastal Monitoring Program. The aims of this paper are to document and interpret the history of beach accretion at Dungeness in four dimensions, length × width (area), × thickness (volume), and time, using a novel combination of historic images, GPR, and Lidar data, quantify rates of sediment accumulation, beach progradation, and assess the impact of 55 years of beach recharge on sediment accumulation downdrift. The Dungeness shoreline is surveyed annually for coastal management, [12,24], but in this study we use freely available remote sensing data and GPR to investigate shoreline morphodynamics and sediment budget on a locally prograding shingle beach.

## 2. Materials and Methods

We use a combination of historic aerial images that are freely available on Google Earth™ and Lidar data to interpret the geomorphology and reconstruct the accretion history at Dungeness. Google Earth Pro™ includes a range of images of Dungeness with aerial photographs dating back to 1940. For this study, we selected images that provide the longest possible record at the highest available resolution; this includes aerial photographs from 1940, 1960, 1990, 2006, and 2019. Lidar data, including 2 m data from 2017 and 2021 as well as 25 cm data from 2017, was downloaded from the Environment Agency National LIDAR Programme—data.gov.uk website on 28 June 2021, and it was analyzed using QGIS and ARC GIS. To assess the error in measurements of the shoreline from remote sensing images, we used fixed points on the ground where the Romney Hythe and Dymchurch

railway crosses the coast roads to check the accuracy of the image registration. Since 2010, image registration has been very good, with displacement between images typically less than 1 m. Images from 1990 to 2010 commonly show offsets of around 2.5 to 3 m between images. The image registration on older aerial photographs from 1960 and 1940 varies between around 4 m and 20 m. The 1940 aerial photographs in Google Earth Pro™ are not perfectly registered with the later images, so we have georectified the position of the 1940 shoreline using fixed points in the landscape, including the road Coast Drive and associated road junctions.

*GPR*

Ground-penetrating radar (GPR) data was collected using a Pulse EKKO pro with 200 MHz antennas spaced 0.5 m apart and arranged in a parallel broadside configuration with a step size of 0.1 m, a time window of 350 ns, and 64 stacks. A total of 1240 m of GPR data was collected along seven profiles perpendicular to the coast, with lines spaced around 500 m apart and seven 50 m long profiles parallel to the coast (Figure 3). Topographic elevations were measured at 1 m intervals along each profile. The GPR data was processed using Pulse EKKO software and the interpretation methodology uses radar facies analysis, as well as radar sequences following the methods described by [25–28] and [15].

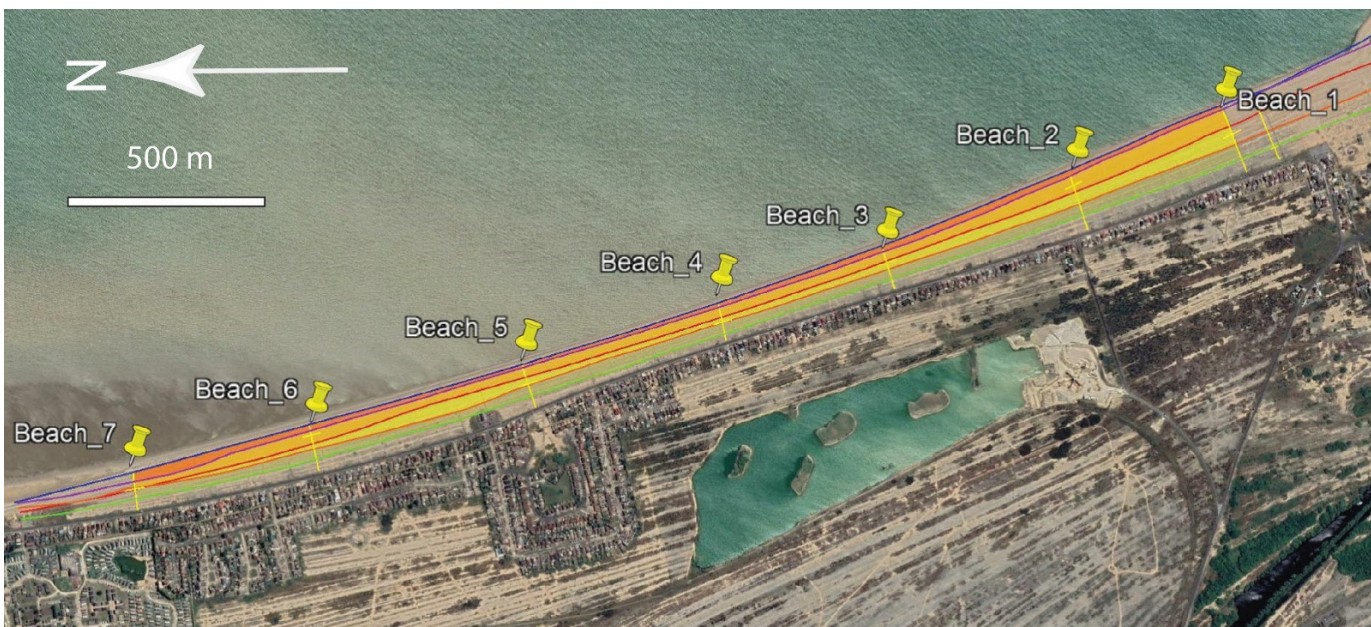

**Figure 3.** Google Earth image of the east-facing beach at Dungeness showing the location of the GPR profiles Beach 1–7 and locations of mapped shorelines from 1940 (green), 1960 (orange), 1990 (red), 2006 (purple), and 2019 (Blue).

## 3. Results

### 3.1. Beach Ridge Patterns

Satellite images and lidar data show beach ridge topography at Dungeness. Along the section studied, we found that the number of beach ridges decreased from south to north (Figure 4).

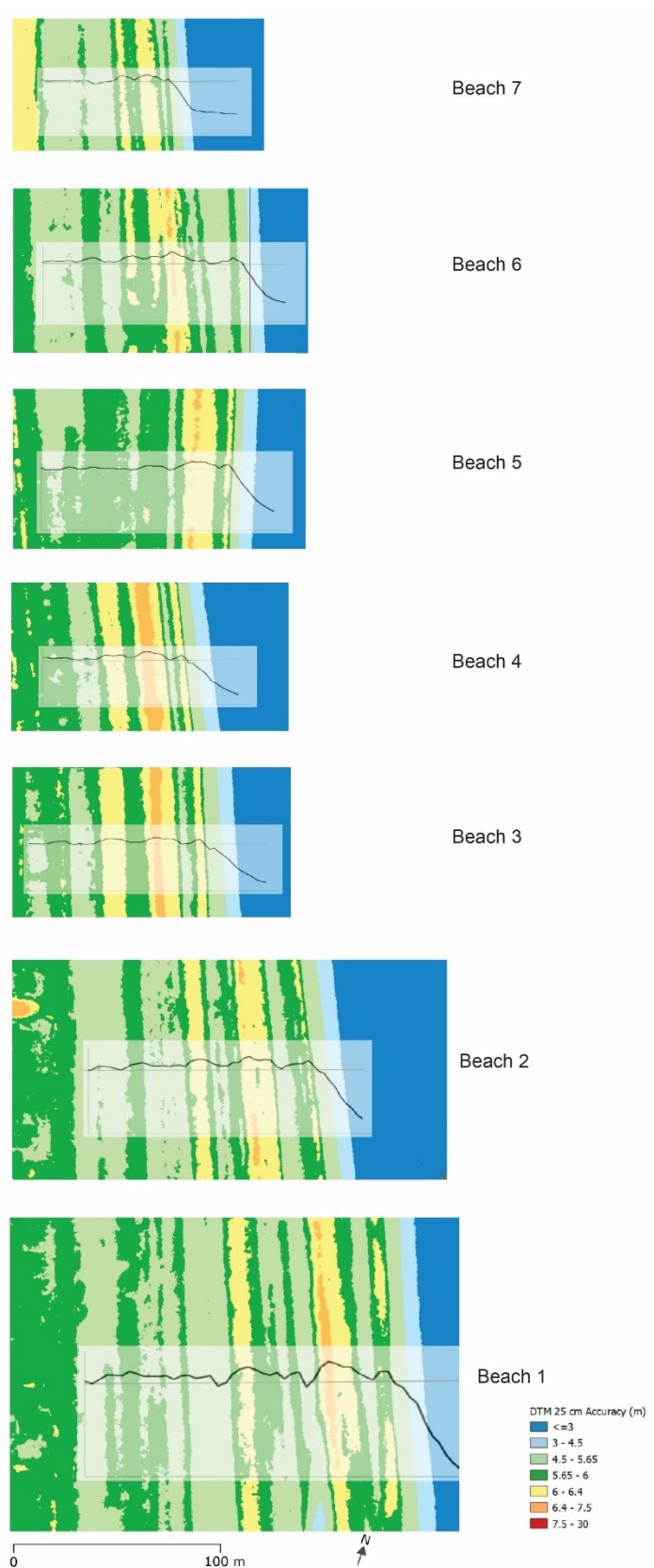

**Figure 4.** The number of beach ridges depicted in dark green, yellow, and orange decreases from south to north along the coast. Elevation maps compiled from 25 cm Lidar data with the location of the GPR survey lines Beach 1–7, see Figures 1 and 3 for locations.

We note three patterns of beach ridge truncation and termination. Terminations are found at both the landward and seaward ends of beach ridges where they onlap against earlier (older) beach ridges. Truncations are formed at the seaward end of beach ridges where they are truncated (eroded) before the deposition of younger ridges that overlap the beach-ridge terminations. Towards the apex of the ness, there are short sections of beach ridge arranged in echelons, parallel rows with successive ridges overlapping, and each ridge terminating against a former shoreline and truncated by a later shoreline (Figure 5a,b). Similar patterns can be found within the salients to the north of the ness and are attributed to accretion from shingle bedforms migrating along and accreting onto the shoreline. At the southern end of the studied section, larger, wider, and higher elevation beach ridges can be traced around 3 km along the coast, but packages of smaller, narrower, and lower elevation beach ridges thin and pinch out towards the north. The higher and wider ridges extend further along the coast than the lower narrower ridges and appear to truncate the smaller lower ridges (Figure 5e,f). This pinch out and truncation of smaller ridges appears to be the main reason for the northward decrease in the number of beach ridges preserved. A third pattern of beach ridges is apparent at the northern end of the studied section, where low-angle sigmoid shaped beach ridges overlap and pinch out towards the north with successive ridges over lapping and extending further along the coast (Figure 5c,d). The sigmoid pattern appears to be due to the movement of sediment along the coast by longshore drift, possibly remnants of the kilometer-scale bedforms observed near the ness, although the accumulation appears to be a on a longer, decadal time scale.

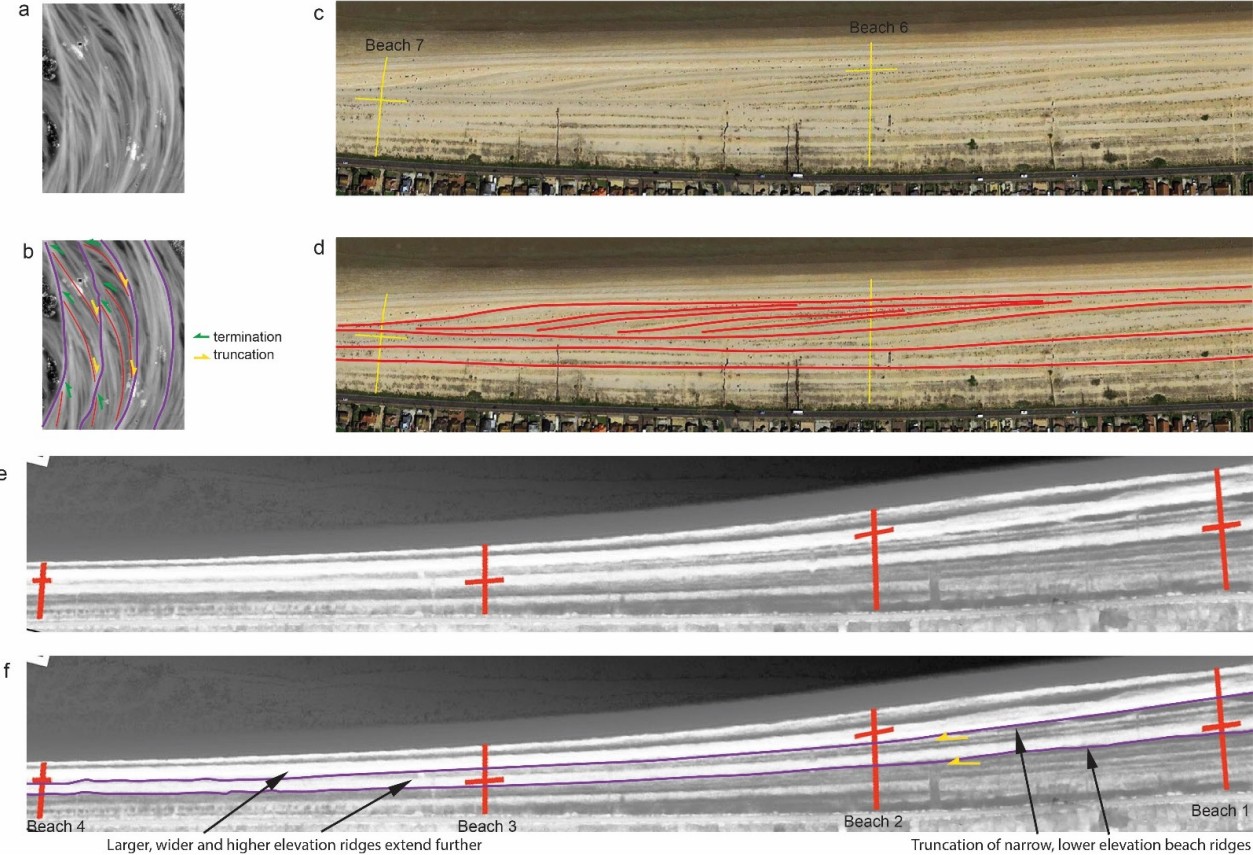

**Figure 5.** Three patterns of beach ridges that record changes in beach ridge preservation along the coast. Panels (**a**,**b**,**e**,**f**) are lidar images, and panel (**c**,**d**) is a satellite image from GoogleEarth™. Panels (**b**,**d**,**f**) are the same as (**a**,**c**,**e**), respectively, with added lines showing interpretation. (**a**,**b**) Short echelon ridges that overlap and terminate against earlier shorelines at the landward end and are truncated by younger shorelines on their seaward ends; (**c**,**d**) low-angle sigmoid oblique beach ridges that thin and overlap towards the north in the direction of longshore drift; (**e**,**f**) truncation of lower elevation and narrower beach ridges by wider, higher elevation beach ridges that extend further along the coast.

Measurements of beach progradation between the shorelines picked from historic images on Google Earth Pro™ are shown graphically in Figure 6, with the data in Table 1. Overall, beach progradation is greater in the south (Beach 1) than it is in the north (Beach 7), Figure 6 and Table 1. The average rate of progradation is 1.85 m/year at Beach 7 to 0.86 m/year at Beach 7 (Table 1). This is a consistent pattern that persists between all of the mapped shorelines between 1940 and 2019. It is worth noting that between 1960 and 1990 the shoreline at profile Beach 7 retreated by 5 m (Figure 6 and Table 1).

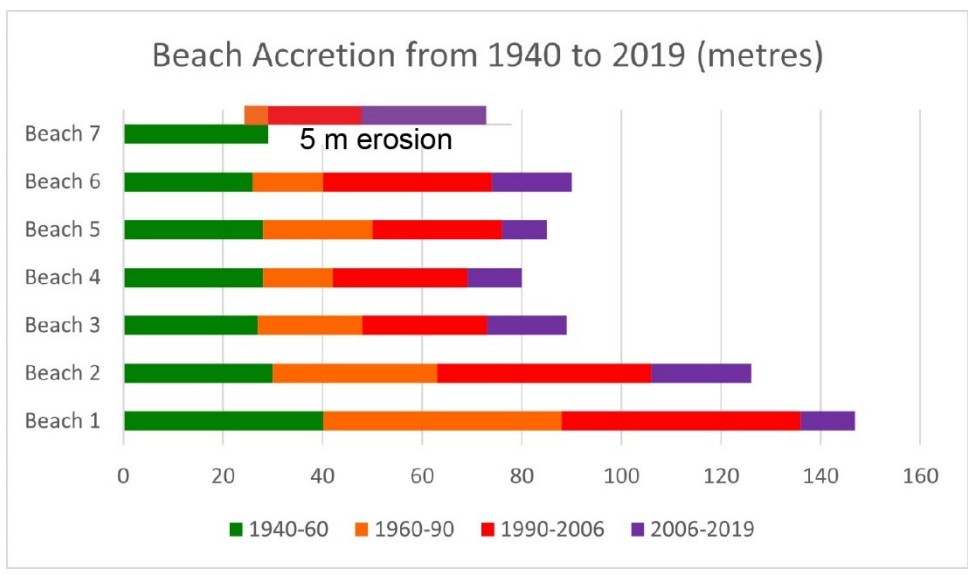

**Figure 6.** Graphic representation of beach accretion at Dungeness between 1940 and 2019. Note the reverse movement at Beach 7 between 1960 and 1990 when the beach was locally eroded, and the shoreline retreated by 5 m. Rates of progradation decrease from south to north along the direction of longshore drift.

**Table 1.** Beach accretion at Dungeness between 1940 and 2019, with average rates of progradation over this time.

| Progradation | Beach 1 | Beach 2 | Beach 3 | Beach 4 | Beach 5 | Beach 6 | Beach 7 |
|---|---|---|---|---|---|---|---|
| 1940–1960 (m) | 40 | 30 | 27 | 28 | 28 | 26 | 29 |
| 1960–1990 (m) | 48 | 33 | 21 | 14 | 22 | 14 | −5 |
| 1990–2006 (m) | 48 | 43 | 25 | 27 | 26 | 34 | 19 |
| 2006–2019 (m) | 11 | 20 | 16 | 11 | 9 | 16 | 25 |
| Total 1940–2019 (m) | 146 | 120 | 89 | 81 | 84 | 90 | 69 |
| Progradation rate 1940–2019 (m/year) | 1.85 | 1.73 | 1.13 | 1.03 | 1.06 | 1.14 | 0.86 |

### 3.2. GPR Interpretation and Radar Facies

At the top of each GPR profile there is a pair of black and white bands that are the direct signal between the transmitter and the reciever. The top band is the 'airwave,' which travels through the air at the speed of light. The second band is the 'groundwave,' which travels through the ground close to the surface at a lower velocity. The velocity of the radar waves in the surface shingle at Dungeness is determined as 0.12 m/ns using field measurements and curve fitting to hyperbolic reflections. Beneath the direct arrivals of the airwave and groundwave are reflections from the depositional layers within the shingle. We used radar facies analysis to identify and interpret GPR reflection patterns. Radar facies are mappable, three-dimensional sedimentary units composed of reflections whose parameters differ from adjacent units [25,26,28]. At Dungeness, we identified four

radar facies in the GPR profiles. Radar facies 1 (RF 1) are planar inclined reflections that dip towards the sea at an angle of around 15° and commonly extend for around 10 m. Radar facies 1 is the dominant reflection pattern on the GPR profile; the example shown in Figure 7C comes from 100–112 m on the GPR profile. RF 1 is commonly found beneath the swales and is interpreted as a beach prograde.

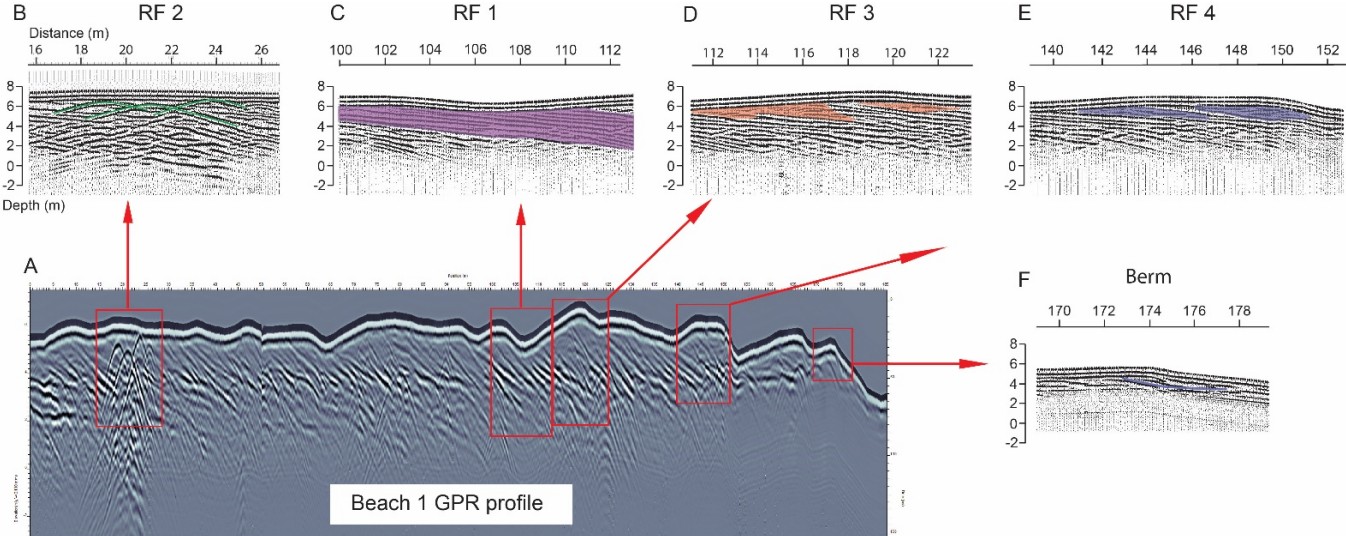

**Figure 7.** Radar facies from GPR profiles across shingle beach ridges at Dungeness. GPR profile of Beach 1 is the most southerly and longest of the seven GPR profiles collected across the beach at Dungeness. (**A**) Shows the profile with considerable vertical exaggeration. Inset boxes (**B–F**) show examples of the radar faces RF 2, RF 1, RF 3, and RF 4, respectively. The dominant reflection pattern is inclined planar reflections (RF 1); this is commonly found beneath the swales and interpreted as beach progrades. RF 2 hyperbolic reflections are interpreted to come from buried objects in the shingle. RF 3 short concave reflections are interpreted as beach berms. RF 4 discontinuous, convex low-amplitude reflections are interpreted as deposits from wave overtopping.

Radar facies 2 (RF 2), high amplitude hyperbolic reflections (Figure 7B), are particularly well developed close to the landward end of the profile between 16 and 26 m. This is between the road that was built in 1935 and the 1940 shingle-sand contact picked from aerial photographs. Parabolic reflections are commonly formed where GPR profiles cross linear features such as pipelines or other buried objects. The location and spacing of the RF 2 hyperbolas on the GPR profile suggest that these reflections come from the remains of anti-tank and anti-personnel defences that were constructed along the beach in the second world war. Such structures can be seen in the 1940 aerial photographs on Google Earth Pro™. Parabolic reflections elsewhere in the profiles may also be from WWII defenses that were bulldozed into the sea at the end of the war, or they could be from other anthropogenic artifacts associated with fishing boats and a range of launch and recovery systems that have been deployed to move fishing boats on and off the beach at Dungeness.

Radar facies 3 (RF 3) comprises short concave reflections. These are typically low amplitude and found within the beach ridges and on the modern active beach. On the modern beach, the concave reflections are associated with berms, and this observation is used to inform the interpretation of the concave reflections as berms preserved within the beach ridges. The examples shown in Figure 7D are from 140–152 m on the GPR profile of Beach 1 (Figure 7D).

Radar facies 4 (RF4) shows short, discontinuous, and convex low-amplitude reflections that down lap in the landward direction and are usually truncated by incline reflections on the seaward side (Figure 6E). RF4 is found within the beach ridges, and the example shown in Figure 7E is from 111 to 123 m on the GPR profile of Beach 1. The convex reflections and landward downlap are interpreted as the product of gravel deposited by overtopping

waves. Similar convex reflection patterns have been recorded from mixed sand and gravel beach ridge deposits on the Sussex coast [29]. In the Suffolk beach ridges surveyed by [29], convex reflections are common, but at Dungeness, RF4 is the least common reflection pattern (Figure 7E).

### 3.3. Radar Stratigraphy

Analysis of the historic images shows that the increase in beach width decreases along the coast from south to north (Table 1 and column 3 of Table 2). In addition, using the change in reflection pattern on the GPR profiles shows that the thickness of the shingle decreases from around 5.2 m in the south (Beach 1) to a thickness of around 4.1 m at Beach 7 in the north (Table 2). Thus, the shingle reduces in thickness as well as reduces in width downdrift from south to north. This is supported by borehole data in [9], which shows gravel deposits thinning from south to north towards Greatstone.

**Table 2.** Cross-section area of shingle beach deposits between 1940 and 2019 derived from GPR profile, the width of the beach accretion between 1940 and 2019, and the average thickness of shingle derived by dividing the cross-sectional area by the width of accretion.

|  | Cross-Section of Beach from GPR (m$^2$) | Increased Width of Beach from Historic Images (m) (1940–2019) | Average Thickness of Shingle at Each Profile Derived by Dividing the Cross-Section Area by the Change in Beach Width (m) |
|---|---|---|---|
| Beach 1 | 758 | 146 | 5.19 |
| Beach 2 | 584 | 120 | 4.85 |
| Beach 3 | 476 | 89 | 5.35 |
| Beach 4 | 402 | 81 | 4.96 |
| Beach 5 | 393 | 84 | 4.68 |
| Beach 6 | 405 | 90 | 4.5 |
| Beach 7 | 278 | 68 | 4.09 |

## 4. Discussion

### 4.1. Definition of the Shoreline on Aerial Photographs and GPR

Defining the shoreline on aerial photographs of Dungeness is difficult due to the limited difference in texture and tone between the foreshore and the backshore, both of which are composed of gravel, as well as the large semi-diurnal tidal range and tidal cycles which change position and elevation daily. In the field, there is a sharp break of slope at the base of the gravel beach where the relatively steep gravel beach meets the lower angle intertidal sand flat (Figure 2), which is a feature of composite gravel beaches [10]. This contact between the gravel foreshore and tidal sand flat, over which the gravel is prograding, can be identified on black and white as well as color aerial photographs and satellite images due to the distinct change in tone between the pale gravel and the darker sand.

The gravel–sand contact used to define the shoreline is visible on the aerial images and can also be identified on GPR profiles due to the change in slope between the relatively steep gravel beach and the lower angle sand flat, where reflections often terminate. As a consequence, we were able to correlate the reflections on the GPR with the position of the shoreline when aerial photographs were taken. This contact is neither the high tide line nor the low tide line, but because the change in composition from gravel to sand is marked by a break in slope as well as a change in tone, we are able to pick the change in tone from aerial photographs and correlate this with a break in slope on the GPR profiles, which occurs around 1 m OD. This enabled us to reconstruct the form and position of the beach, since the earliest available aerial photographs were taken in 1940, and to estimate the volumes of shingle that have accumulated along the studied section of coast since 1940.

Errors in image registration are typically between 2.5 and 3 m, which is around 3% of the measured change in beach width between 1940 and 2019.

### 4.2. Beach Ridge Morphodynamics and Radar Facies

There are a relatively small number of GPR studies of mixed sand and shingle beaches, and the results tend to show two contrasting reflection patterns. GPR surveys of beach ridges in British Colombia show beach face progradational facies with limited 'beach cap' facies [17]. Seaward inclined beach face progrades, from beach ridges on the Island of Anholt, are reported by [30], and this reflection pattern was confirmed by [31]. In contrast, GPR profiles across mixed sand and gravel beach ridges in Canada described by [32] show lens-like geometries and convex reflections with onlapping landward terminations attributed to vertical accretion and overwash on the beach ridges. Lens-like geometries and low-angle landward dipping reflection patterns with seaward dipping bounding surfaces within sand and gravel beach ridges on the Suffolk coast, UK are also reported by [29]. These are attributed to landward migration of multiple berm ridges with overtop and overwash deposits during storm events to form beach ridges [29].

The two reflection patterns that appear to correspond with the two types of beach ridges, Mode I and Mode II, described by [33]. Mode I forms comprise seaward dipping swash units and are associated with shore-normal sediment movement (Carter 1986). Mode II beach ridges are dominated by landward dipping units associated with spillover and washover. Mode II is associated with longshore drift [33], which presents a conundrum, because Dungeness is dominated by longshore drift but shows the seaward inclined beachface reflection pattern expected for shore normal, rather than longshore, sediment transport.

### 4.3. Estimates of Shingle Accumulation

The total volume of shingle accumulation between 1940 and 2019 along the section of coast studied is 1,355,660 m$^3$, giving an annual rate of accumulation of 17,160 m$^3$/year (Table 3), values are expressed to the nearest 10 m. The corresponding figures for rates of sediment accumulation from coastal monitoring between 2003 and 2012, from annual ground-based GPS measurements reported for sub cells RS4 and RS5, is 22,175 m$^3$/year [12]. The section described here is 500 m shorter than sub cells RS4 and RS5, and the difference in the length of the shoreline measured, as well as differences in the way that the thickness of the beach are calculated, can account for most of the difference between the values that we have derived from remote sensing data and the values measured from shoreline monitoring data. Alternatively, short term erosive events associated with the passage of salient produced by high-angle waves could interrupt and locally reverse accretion, resulting in lower rates over a decadal timescale.

**Table 3.** Calculations of the area of accretion from Google Earth Pro™, annual rates of accretion, progadation rate, shingle beach volume, average annual volume of beach accretion, mass of shingle, and rates of beach accretion. Values have been rounded up and down to the nearest 10 m.

| | Area of Accretion (m$^2$) | Accretion Rate (m$^2$/year) | Progradation Rate (m/year) | Volume of Beach Accretion (m$^3$) | Average Annual Volume of Beach Accretion (m$^3$/year) | Mass of Shingle (Tonnes) | Rate of Beach Accretion (Tonnes/year) |
|---|---|---|---|---|---|---|---|
| 1940–1960 | 83,000 | 4150 | 1.40 | 398,390 | 19,920 | 717,090 | 35,860 |
| 1960–1990 | 66,730 | 2220 | 0.75 | 318,580 | 10,620 | 573,440 | 19,120 |
| 1990–2006 | 89,500 | 5590 | 1.88 | 429,610 | 26,850 | 773,310 | 48,330 |
| 2006–2019 | 43,200 | 3320 | 1.12 | 207,360 | 15,950 | 373,240 | 28,710 |
| 1940–2019 | 282,430 | 3580 | 1.20 | 1,355,660 | 17,160 | 2,440,190 | 30,890 |

### 4.4. Impact of Beach Recharge

The density of shingle at Dungeness is 1800 kg/m$^3$ [1]. We calculate that just over 2 million tons of shingle has accumulated along the beach studied between 1940 and 2019. This gives an average annual rate of accumulation of 26,512 t/year between 1940 and 2019, with rates varying from 30,853 t/year between (1940–1960) to 21,691 t/year between 1960–1990. If this sediment were spread evenly along the coast, it would amount to 8.9 t/m/year. However, the sediment is not spread evenly along the coast, and the rates of accumulation at the southern end of the section studied are twice those in the north due to a gradual northward decrease in longshore drift. From a temporal perspective, the rates of sediment accumulation are highest between 1940 and 1960, and they are lowest between 1908 and 1939. The sediment accumulation rate also decreases between 1960 and 1990 (Figure 8B). This could be taken as an indication that construction of the nuclear power stations in the 1960s and the inception of the beach recharge scheme in 1965 did have an effect on the rates of shingle accumulation on the eastern shore at Dungeness. However, the amount of shingle extracted annually, around 30,000 m$^3$/year, is equivalent to two year's worth of sediment accumulation along the section of coast studied. In addition, this sediment is returned to the beach on the updrift side of the power stations and is therefore likely to have a short-term, rather than long-term, effect. Furthermore, it should be noted that the 1960s was a decade with relatively few coastal floods in southern and southwest England [34], likely due to a decrease in storms associated with a negative NAO [34] (Figure 8).

### 4.5. Storms

The formation of beach ridges on coarse grained, gravel, or shingle beaches is generally attributed to storm events [5]. This is because coarse-grained, shingle beaches have a high permeability, enabling water to drain into the shingle, reducing backwash and effectively trapping coarse grains on the ridge crest. Landward dips indicative of overtopping and washover are rare in the beach ridges at Dungeness, possibly only occurring when the beach becomes saturated, and infiltration is reduced. Coastal flood frequencies in Britain since the 1780s have been compiled by [34] in three sectors, one of which, the south and southwest sector, includes Dungeness. A relatively low number of floods is recorded on the south coast for the 19th century, from the 1890s to the 1940s flood frequencies increased [34]. The 1960s show a decrease in the number of reported floods, which is associated with a negative NAO (Figure 8). Recorded floods increased in the 1970s and decrease towards 2000 [34]. It should be noted that the number of reported floods can be affected by cultural factors, particularly expansion of coastal towns, creating an increased flood risk, while later improvements in flood defenses reduced vulnerability and flooding [34]. A reduction in coastal flooding from southwesterly storms provides an alternative explanation for a potential decrease in erosion from the southwest shore and consequent decrease in sediment accumulation on the east facing shore. In addition, the rates of sediment accumulation since 1990 have returned to values that are similar to those that pertained between 1940 and 1960, suggesting that there is no obvious long-term trend in the annual rates of accretion before and after the construction of the nuclear power stations at Dungeness. If anything, the rates of accretion appear to have increased slightly during the past century (Figure 8), possibly a result of increased erosion on the updrift, southwest-facing shore. However, the duration of the time gaps between freely available aerial survey data, typically 20 to 30 years, precludes more detailed analysis of the rates of change. It is suggested here that analysis of the erosion rates along the southwest-facing shore would help to resolve this question because that might provide an independent assessment of the potential sediment supply due to erosion.

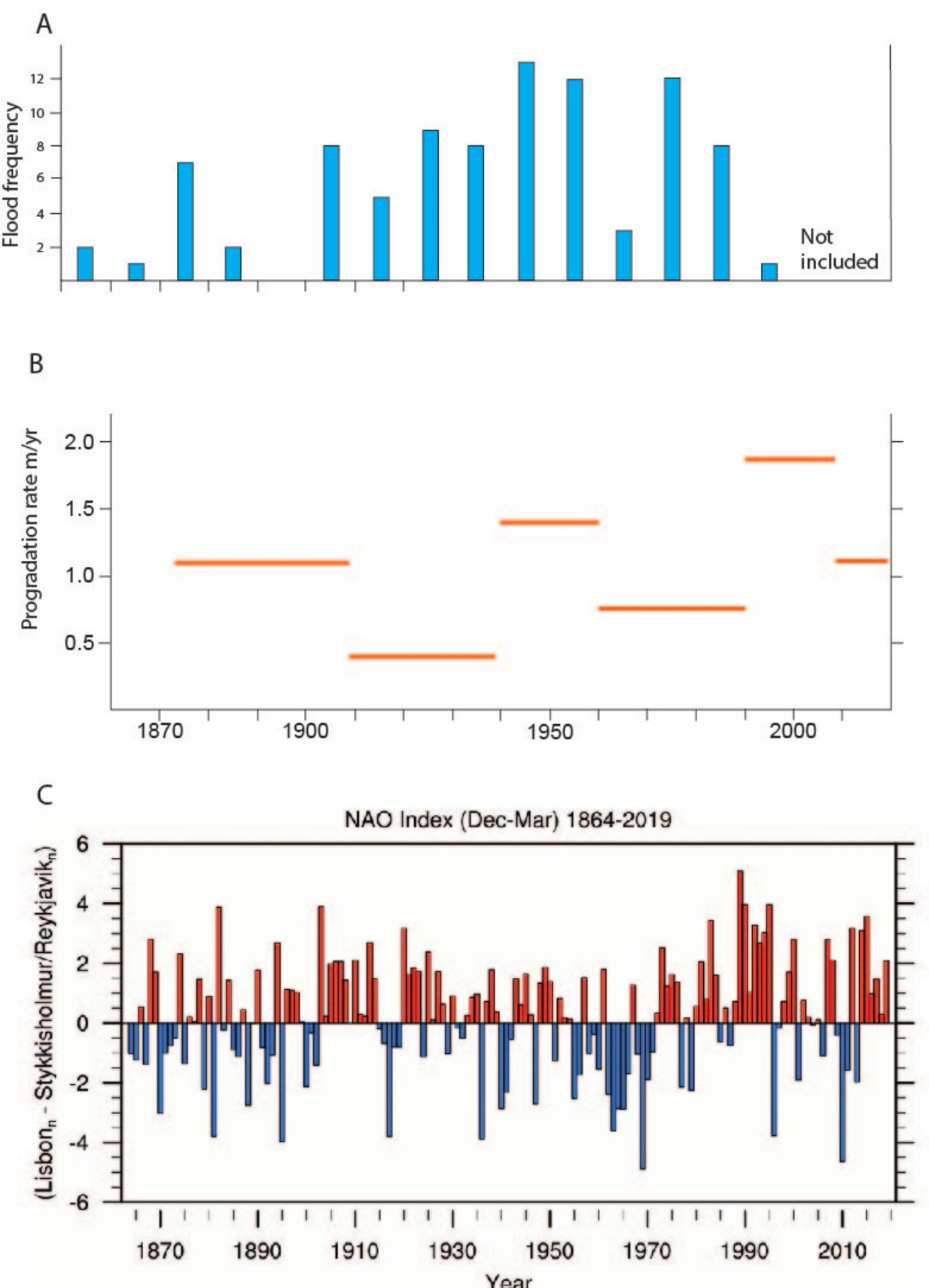

**Figure 8.** (**A**) Flood frequencies for southwest England since the 1780s, compiled by [34]. (**B**) Rates of progradation determined in this study. (**C**) Hurrell North Atlantic Oscillation (NAO) winter (December–March) station-based index from [35].

### 4.6. Confirmation of Lewis's Field Observations

Lewis [2] wrote a seminal paper on the formation of Dungeness foreland and visited the ness in 1929 and 1931. His observations of accretion around the point at Dungeness are recorded in a sketch (Figure 9A). Lewis [2] noted that the orientation of the beach with respect to the prevailing south-west waves was critical in determining the rates of longshore drift and beach accretion at the southern end of Dungeness. He noted that 'when the bend was slight it had little effect upon the rate of drift, but as it (Dungeness point) became sharper the drift along the leeward shore was lessened. For the prevalent south-west waves which cause this drift are so weakened in swinging round the bend that in spite of their great obliquity they are unable to drift material northwards from the Ness as rapidly as they bring it along the southern shore. This results in large supplies of shingle accumulating immediately around the point, which in turn is built into ridges overlapping the point by the south-west waves, thus causing the Ness to advance seawards' [2] (p. 3190). Lewis termed the projections on the ness 'salients,' while recent modelling shows that such features can be formed by high-angle waves [24].

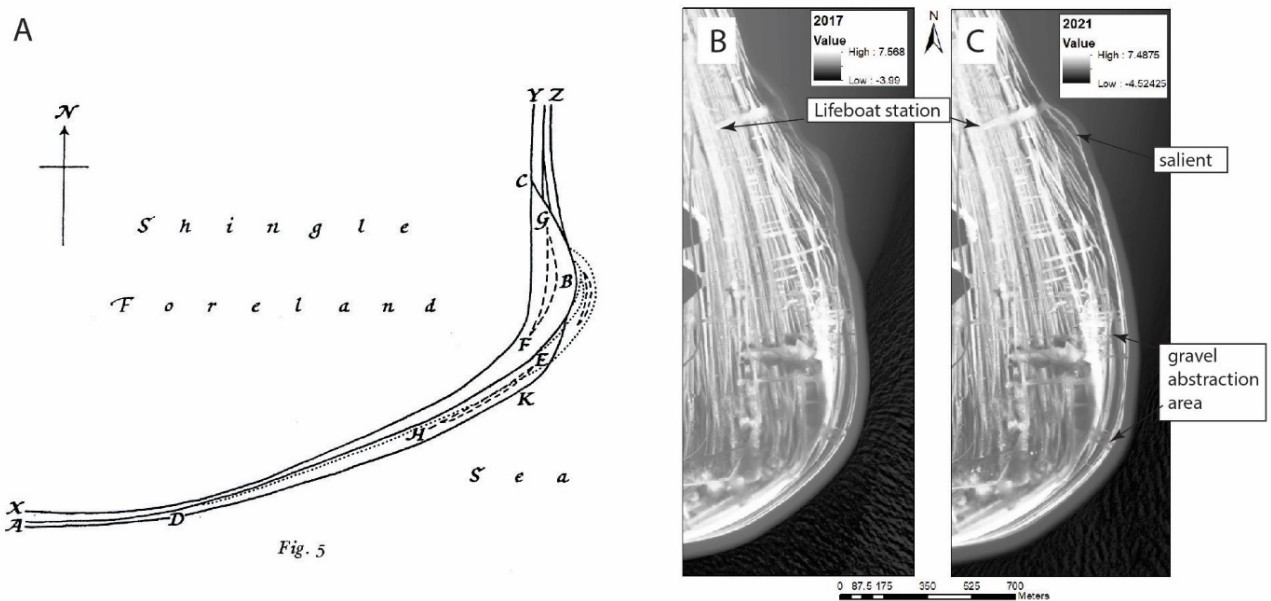

**Figure 9.** Salients close to the apex of Dungeness; (**A**) as illustrated by Lewis (1932). (**B**) Lidar data from 2017 and (**C**) Lidar data from 2021 showing the recent development of salients and echelon beach ridge formation.

The same processes and resulting accretion patterns can be seen occurring today using Lidar data collected in 2017 and 2021 (Figure 9B,C). The 2017 survey (Figure 9B) shows two small salients of around 450 m wavelength that are migrating north [24]. By 2021, the northern salient has become reduced, and the southern salient has continued to migrate north (Figure 9C). Topographic survey data from 2010 to 2016 [24] (Figure 3) shows that the salients had formed in 2013–2014 in response to high-angle waves [24]. These were incorrectly described as sand waves by [24] because they are primarily composed of gravel, and we prefer the non-genetic term salient following the precedent set by [2]. It is notable that these features formed during the winter of 2013 when wave modelling by [24] showed a dominance of high-angle waves from the south-west. We suggest that they are formed of sediment eroded from the swash aligned south-west side of Dungeness and deposited at the Ness as the waves refract around the point, creating spits at the apex of the ness that become recurved as the dominant south-west waves are refracted around the ness and then reattached to the shore. The reattachment of spits creates depressions, known locally as pits, that are noted by [4]. The northward movement of shingle continues along the east side of the ness. This is in contrast to [2], who stated that 'the south-westerly and southerly waves are very much reduced in size after swinging round to reach this lee

shore, and can therefore be neglected' [2], p. 21. Between Lydd on Sea and Greatstone, northward longshore drift deceases and converges with westward longshore drift [4,12]. This northward decrease in longshore drift is accompanied by a decrease in the thickness and the width of the shingle beach at Dungeness, as well as a decrease in the number of preserved beach ridges. This northward transport continues in 2020 and 2021.

*4.7. Beach Ridge Truncation*

The shingle ridges of Dungeness record the contemporary shape of the shoreline when they were formed, and they consequently record the evolution of Dungeness. The beach ridges are not completely parallel and can be divided into beach ridge sets. Our observations show that there are a greater number of beach ridges on the southern profile, 11 on Beach 1, and fewer beach ridges at the northern end, seven at Beach 7 (Figure 4). The formation, or more accurately the preservation, of different numbers of beach ridges within the same time period on the same stretch of coast has important implications for the use of beach ridges in reconstructing coastal evolution and estimation of high magnitude events such as storms, e.g., [20,22,36–39]. To the north of the salient, the beach is eroding, and this creates a shallow embayment that is infilled by the reattached spit-creating beach ridges at an oblique angle to the coastline. Examples of packages of oblique beach ridges are preserved at the Ness (Figure 5A,B). Besides the formation of oblique ridges, it is notable that the beach is eroded ahead of the salients. This erosion has the potential to remove earlier beach ridges. Erosion downdrift of salient has the potential to reduce the value of beach ridges as records of storm events because the erosion could remove earlier beach ridges creating gaps in the beach ridge record. Despite this, it does appear that larger, wider, and higher ridges, potentially formed during storm events with greater wave run-up and overtopping, have a higher preservation potential than narrower, lower amplitude ridges that have been observed to pinch out alongshore and are truncated by the larger ridges.

## 5. Conclusions

Aerial photographs, Lidar data, and satellite images have been used to map changes in the shoreline locations along a prograding shingle shoreline at Dungeness from 1940 to 2019. Ground-penetrating radar (GPR) reveals the internal structure of the beach ridges and the thickness of shingle on this composite gravel beach. The beach ridges are underlain by seaward inclined reflections from beachface progrades, and the thickness of the shingle decreases from south to north. By combining mapping of the planform areas of accretion with measurements of beach thickness, we have calculated volumes of sediment accumulation over the past 79 years (1940–2019). The width and the thickness of shingle decreases from south to north, which is the direction of longshore drift. Mapping of sediment accretion from beach ridges at Dungeness tend to confirm earlier observations and interpretations of the formation of Dungeness cuspate foreland made by [2]. We disagree over the accretion of beach ridges on the east facing shore to the north of the Ness where [2] considered that the longshore drift from southwest waves could be neglected. The northward decrease in the width and thickness of the beach ridges is testament to be continued but by reducing northward longshore drift along the eastern side of Dungeness. The northward movement of sediment along the coast, and resulting overlap of beach ridge accretion surfaces, and northward decrease in rates of accretion is best explained by northward longshore drift. While erosion downdrift from salients impacts the preservation of beach ridges, this does not exclude easterly storm waves as geomorphic agents in construction of beach ridges on the east side of Dungeness, but we suggest that the south-west waves continue to play a significant role in the sediment transport along this drift aligned shore. The preservation of different numbers of beach ridges along the coast within a known time period raises questions for the reconstruction of storm records from beach ridge sequences. Estimates of sediment accumulation using GPR profiles and freely available remote sensing data agree well with measurements of sediment accumulation from shoreline monitoring data from annual ground-based GPS surveys. There is a decrease in the rates of accretion between

1960 and 1990, which is a possible consequence of the beach recharge schemes put in place to protect the nuclear power station, but it is also worth noting that a decrease in storms during the 1960s could have had a similar effect by reducing sediment erosion on the southwest facing shore. It should be noted that since 1990, beach accretion on the east side of Dungeness, at Lydd on Sea and approaching Greatstone, has returned to values similar to those before the beach recharge scheme started, suggesting that while there might have been a short-term impact this has not persisted. We conclude that the beach recharge scheme used to protect the two nuclear power stations at Dungeness does not appear to have had a negative impact on the beach downdrift. A major limitation of this study is the granularity of the historic images, the periods between aerial surveys that prevents more detailed analysis of the rates, and timing of beach accretion.

**Author Contributions:** Project conceived by C.S.B. who supervised L.B. and M.I. M.I. and L.B. collected the GPR data and processed the Lidar data. C.S.B. wrote the manuscript. All authors have read and agreed to the published version of the manuscript.

**Funding:** L.B. is funded by London NERC DTP grant NE/L002485/1, and MI was funded by C N Yang Scholars Program, NTU.

**Institutional Review Board Statement:** Not applicable.

**Informed Consent Statement:** Not applicable.

**Data Availability Statement:** GPR and topographic data is archived at the Birkbeck Data Repository BiRD https://doi.org/10.18743/DATA.00160 (accessed on 22 September 2021).

**Acknowledgments:** We would like to thank the warden and landowners at Dungeness for permission to work there.

**Conflicts of Interest:** The authors declare no conflict of interest.

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
