# Peer review of "Four-Dimensional Investigation of Gravel Beach Ridge Accretion and 50 Years of Beach Recharge at Dungeness, UK, Using Historic Images, GPR and Lidar (HIGL)"

_applsci, doi:10.3390/app112110219_

Round 1

Reviewer 1 Report

Four-dimensional investigation of gravel beach ridge accretion and 50 years of beach recharge at Dungeness, UK, using historic images, GPR and Lidar (HIGL) 

by Bristowet al.

General comments

In my opinion, the manuscript is well written and the aims are clear. The state-of-the-art is well delineated, the methodology is sound. The conclusions are supported by the results.

I suggest minor improvements.

Suggestions

1) The Conclusions section should discuss the implications for the protection of the nuclear power plants, providing suggestions to authorities and stakeholders. Furthermore, the Section should include the limitations of the research and future outlooks;

2) Line 135. Indicate the date when the LIDAR data was downloaded.

Author Response

We thank the anonymous reviewer 1 for their comments.

  1. Conclusions. New sentence added to state: 'The beach recharge scheme used to protect the nuclear power stations at Dungeness does not appear to have a negative impact on the beach downdrift. A major limitation of this study is the granularity of the historic images.'
  2. Date when Lidar data was downloaded has been added to the text (line 123).

Reviewer 2 Report

Around line 128: You used old photographs to establish the coastline in previous years. But which coastline did you use, HW line, LW line? And how to know that from a photograph  when you don’t know the exact time of the picture? Lidar data are probably related not to the tide level, but to Ordnance Level. How did  you handle this differences?

Line 165: When I have understood the picture correctly, the colours  show the hight of the beach with respect to a datum. GPR data should show also layers below the surface. I would expect that just seaward of the top of a beach ridge crest there would be a slope recognisable in GPR data of the ridge of that moment. Is that correct?  I don’t find any information on this in the paper. Most probably I should have been able to find a clue in figure 7, bet for me this need somewhat more explanation.

Th paragraph starting in line 169 explaining figure  5 is not really clear to me.  Figure A and B are pictures in X_Y direction? If yes, where are they located?

Is it correct that figures A, C and D are identical to B, D and E, but only without the explaining coloured lines?

Does the GPR show at the slope of the ridge below she present day surface also the differences between termination and truncation?

For readers not too familiar with GPR the term “radar facies” requires some more explanation than only line 209 and 210.

Table 2: Is my assumption correct that you measured from GPR column 2 and 3, and that you calculated column 4 from the data of column 2 and 3 ?  In that case it is no surprise that the average thickness is decreasing. This follows simply from the fact that it are triangles. For morphological reasons the seaward slope of the shingle is more or less constant. Material is transported from south to north, and on its way to the north material is deposited, until the transport of shingle is simply zero (around Seaview Road) because there the orientation of the coast changes from 275 degrees to 260 degrees. It seems that at this coast the dominant wave comes from 260 degrees, and therefore causing no shingle transport north of Seaview Road. South of the Lifeboat station there is more supply, but part of this is artificially brought back to a point west of the power station.

In line 308 you present an accretion with an accuracy of  1 m3 . Of course this accuracy is a numerical artefact.  Please mention this.

Section 4.2 – 4.5 are based on analysis of standard shoreline observations over time. It is not clear to me what the additional GPR data add to a simple visual map analysis when one is interested in the accumulation data only.

The findings in section 4.5 (second one,  confirmation) seems logical. But I do not see any link with the results from the lidar and GPR data as presented in this paper.

Round 2

Reviewer 2 Report

No further comments